# Occurrence Characterization and Contamination Risk Evaluation of Microplastics in Hefei's Urban Wastewater Treatment Plant

**Xiangwu Meng [1], Teng Bao [1,2,3,*], Lei Hong [1,2,3] and Ke Wu [1,2,3,*]**

1   School of Biology, Food and Environment, Hefei University, Hefei 230601, China
2   Hefei Institute of Environmental Engineering, Hefei University, Hefei 230601, China
3   Collaborative Innovation Center for Environmental Pollution Control and Ecological Restoration of Anhui Province, Hefei 230601, China
*   Correspondence: tengbao1222@sina.com (T.B.); wuke@hfuu.edu.cn (K.W.)

**Abstract:** As one of the primary nodes in the flow of micro-plastics (MPs) in the environment, it is critical to examine and assess the Sewage Treatment, occurrence, and removal of MPs in waste treatment plant (WWTP). This research explored the shape, size, and composition of MPs at various stages of the WWTP process in the south of the city of Hefei, China, in dry and rainy weather conditions, as well as the removal effectiveness of MPs in a three-stage process. The collected MPs were quantitatively and qualitatively examined using an Osmosis electron microscope and micro-FTIR. The pollution risk of MPs in WWTP was assessed using the EU classification, labelling and packaging (CLP) standard and the pollution load index (PLI). The findings revealed that the average abundance of fibrous MPs was greatest in WWTP sewage and sludge, 49.3% and 39.7% in dry weather, and 50.1% and 43.2% in rainy weather, respectively. The average distribution of MPs in the 0–500 μm range was highest in WWTP wastewater and sludge, 64.9% and 60.4% in dry weather and 67.9% and 69.0% in rainy weather, respectively. Finally, the overall removal rate was 87.7% and 83.5%. At the same time, it has been demonstrated that MPs with varied compositions are strongly tied to human activities, and environmental conditions (such as rainy weather) also influence their source. In both dry and wet weather, the amount of polymers and the risk score were linked to the pollution risk of MPs in WWTP. In wet weather, the MPS pollution index was more variable. The pollution indices of MPs in row water and tail water were 2.40 and 2.46, respectively, which were heavily contaminated, and 1.0 and 1.2, which were moderately polluted. MPs in dewatered sludge had severely polluted indexes of 3.5 and 3.4, respectively. As a result, there is still MPs efflux or buildup in sludge during and after the WWTP process, which presents an ecological contamination concern.

**Keywords:** microplastics; wastewater treatment plant; dry and rainy weather; occurrence characteristic; pollution risk evaluation

## 1. Introduction

Plastic is an organic polymer material that is extensively utilized in insulation, metal substitution, packaging, clothing, and other applications. By 2020, China's annual production of plastic products had reached 76.032 million tonnes, with 74.1 million tonnes of garbage, 30% of which was recycled, 32% disposed of in landfills, 31% burned, and roughly 7% lost. In Europe, the total quantity of recycled plastics is 4.6 million tonnes per year, with a 35% recycling rate [1,2]. Microplastics, MPs are a new category of pollutant described as plastic particles with a particle size of less than 5 mm [3]. Primary plastics and secondary plastics are the major sources: primary plastics are mostly formed by grinding particles and plastic beads included in industrial raw materials and cosmetics, as well as the loss of synthetic fibers in clothing caused by washing, etc. [4,5]. Secondary plastics are primarily created by photocatalysis, weathering, embrittlement, and cracking

of large abandoned plastics in the presence of light, water, and wind, which leads to the development of smaller microplastic particles [6,7].

Early MP research primarily focused on the distribution and movement of marine, lake, and river MPs, as well as the source, abundance, and biological toxicity of MPs. It has been documented that plastics distributed in the ocean are more readily broken down into microplastics by photocatalysis and water flow, and that tidal movement can reflow microplastics deposited in the ocean back into the freshwater environment [8]. MPs in the environment have been discovered to operate as a carrier for microorganisms such as pathogens [9], adsorbing organic substances (antibiotics, phthalate, Polycyclic aro-matic hydrocarbon, etc.), and heavy metals [10–13]. Microorganisms can colonise the surface of MPs and create biofilms, transferring a range of bacteria, including dangerous ones, to new environments [14]. Its potential ecotoxicity can be transmitted through the food chain [15–17], is easily acquired by zooplankton and higher animals such as humans, and accumulates in organisms, disrupting the flow of energy within organisms and posing a growth threat [18], even causing biological death [19], with the potential for irreversible risk. Furthermore, MPs has been found in human blood and faeces, as well as in the maternal placenta [20–23]. The primary sources of MPs in the body include daily consumption of table salt, bottled water, tap water, and seafood [24–26], whereas air-exposed microplastics may also be absorbed by inhaling airflow [27,28]. With the potential for ecotoxicity and far-reaching effects on the natural environment and human health, identifying pollution sources is critical for evaluating the pollution risks of MPs and establishing mitigation solutions. As a consequence, the Sewage Treatment monitoring of the whole MPs process was employed as a pilot study to determine the contamination risk.

The primary point at which MPs reach the natural environment from the urban water system is sewage treatment [29,30]. Sherri et al. examined 90 samples from 17 phases of the US sewage treatment process and discovered that up to 4 million MPs particles are still discharged into the natural environment per day following regular sewage treatment [31]. Xu et al. evaluated 11 sewage treatment plants in Changzhou, China, and discovered that the average concentration of MPs in the influent and effluent was $196.00 \pm 11.89$ n/L and $9.04 \pm 1.12$ n/L, respectively. The average removal rate of MPs was over 90%, with the highest percentage being 97.15% [32]. Zhang et al. assessed an MPs removal rate of 93.7% in the entering and departing water of the Turkish Sewage Treatment [33]. However, most of the MPs removed by the sewage treatment process are transferred and stored in sludge, and Esther et al., in their study of the WWTP in Vancouver, Canada, found that $1.76 \pm 0.31 \times 10^{12}$ MPs accumulate in the sewage treatment each year, of which, $(1.28 \pm 0.54) \times 10^{12}$ MPs settled into the primary sludge, $(0.36 \pm 0.22) \times 10^{12}$ MPs into the secondary sludge, and $(0.03 \pm 0.01) \times 10^{12}$ MPs were released into the natural environment [34]. Furthermore, Kay et al. reported that MPs abundance in river basins may increase with atmospheric deposition or agricultural soil infiltration in a study of MPs hosted upstream and downstream in six Sewage Treatment of distinct river basins [35]. Plastic film, microfibers, and inappropriately disposed of waste plastics used in agriculture and industry, on the other hand, degrade into fine plastic particles by a sequence of synergistic photocatalysis and physicochemical degradation processes [6,7,36]. These particles are extensively spread in the urban surface environment and in atmospheric flotsam, and may enter the sewage network when rainfall washes them away [37], which may then be transported to the Sewage Treatment, resulting in an increased treatment load [12,38,39]. The load of MPs, a frequent Persistent organic pollutant in WWTP such as Polycyclic aromatic hydrocarbon, would grow with increased rainfall [40], which would spread the degree of environmental contamination to some extent.

There is currently limited research on the evaluation of MPs pollution risk in urban WWTPs in Hefei City. This paper investigated the form, size, and composition features of MPs in each typical process step of a WWTP under two weather conditions, dry and rain, as well as the removal effectiveness of MPs in the three-stage treatment stage. The pollution risk of MPs in WWTP is analysed using the EU classification, labelling and packaging

(CLP) standard and the PLI Pollution load index model. This report serves as a reference for future MPs reductions in China's inland cities' WWTP.

## 2. Materials and Experimental Methods

### 2.1. Sample Sites

In this study, samples of sewage and sludge were collected from Hefei, China's Sewage Treatment in the city's southern region. The WWTP is equipped with an (Anaerobic-AnoxicOxic process, A$^2$O) with a daily capacity of 100,000 tonnes. The purified sewage is disposed of in the 15li River. Due to the fact that water quality and quantity indicators of the Sewage Treatment may vary depending on the weather, samples were obtained in July 2022 under dry (marked D) and rainy (marked R) conditions. Figure 1 and Table 1 showcases grid intakes (Row water, labeled D1 and R1), grid outlets (labeled D2 and R2), aeration grit chamber outlets (labeled D3 and R3), oxidation ditch outlets (labeled D4 and R4), in the secondary clarifier (D5 and R5), the outlet of the secondary clarifier (D6 and R6), the outlet of the denitrification deep bed filter (Tail water, D7 and R7). The sludge samples are dewatered sludge from the sludge pumping station (D8 and R8).

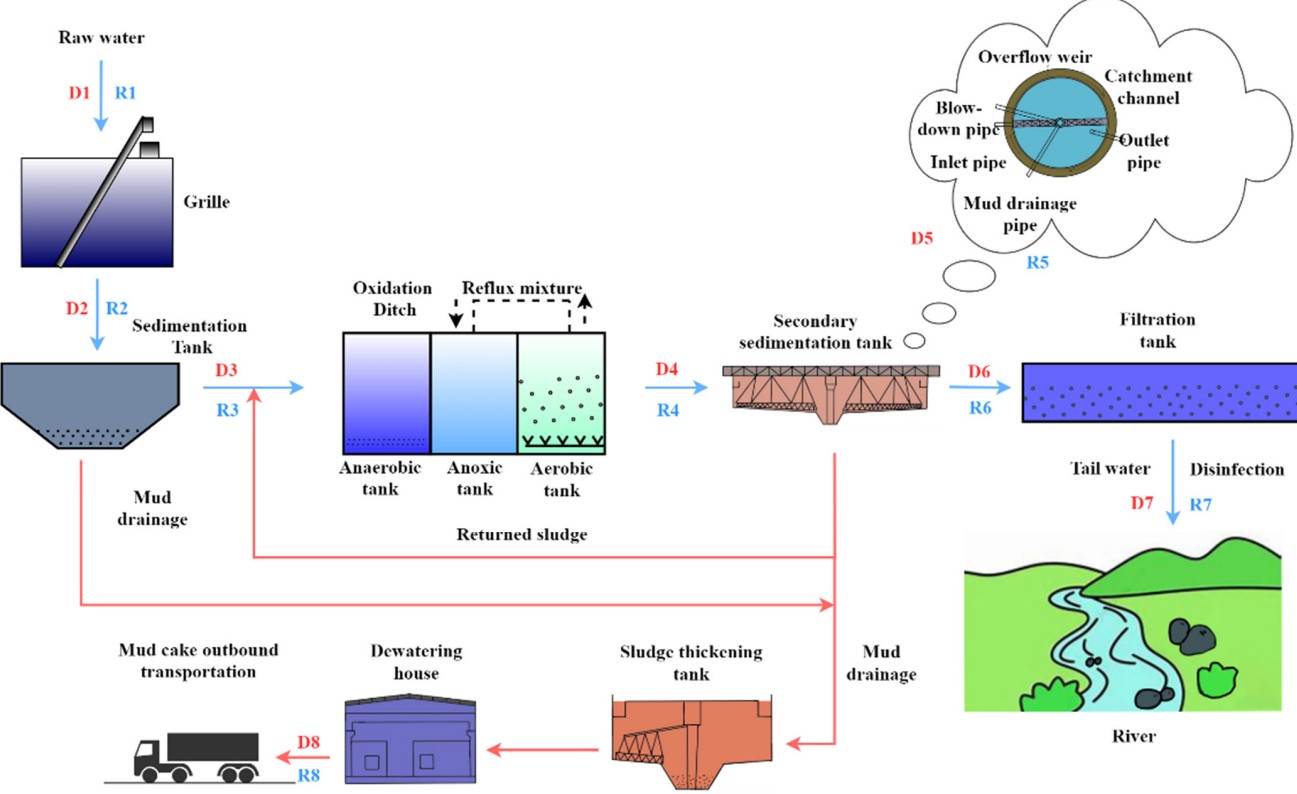

**Figure 1.** Wastewater treatment plant process flow design.

### 2.2. Sample Methods

The quantitative collection of MPs at the Sewage Treatment sample point is performed using a peristaltic pump with a custom stainless steel hopper connected to the pump's intake (20 mm inner diameter of the inlet) and a mesh (5 mm mesh aperture) covering the funnel mouth. Place the pump input pipe with a custom hopper 30 cm below the surface of the sewage for collecting samples, and store the sewage from the pump outlet pipe (10 L of sewage from each sample point) in 15 L stainless steel drums. For sludge samples, the dewatered sludge from the sludge pumping station (75–79% water content) was collected by wrapping 1 Kg samples of sludge in aluminium foil paper and placing them on a conveyor belt. Collect 1 Kg of sludge samples from three distinct points on the conveyor belt, combine the samples, and store them in a 5 L stainless steel drum. In addition, the Sewage Treatment

flow varies in real-time, and the temporal disparities between certain samples generate a data flow, resulting in variations in MPs' features(Table 2). Therefore, samples were collected at each sampling station at three separate times during the day (08:30–11:30, 13:30–16:30, 18:30–21:30), with Sewage Treatment flow sufficient for peak phases. After transferring the sample to the container and rinsing the collector with deionized water, the next collection activity may begin. The edges of the container's top lid are covered with aluminium foil before being returned to the laboratory. For testing and analysis, the container is placed in a refrigerated environment at 4 °C.

**Table 1.** WWTP process section layout design and description.

| Sample Points D (Dry Weather), R (Rainy Weather) | Layout Design Description |
|---|---|
| Raw water or Grille Front (D1, R1) | Initial wastewater from WWTP. Sewage samples are collected at the grate. |
| After grille (D2, R2) | The first physical interception processing. Sewage samples are collected behind the grate. |
| After sedimentation Tank (D3, R3) | Gravity deposition of high-density contaminated impurities separates low density suspended matter to the next process. The sewage samples were collected after sand-settling. |
| After oxidation Ditch (D4, R4) | (Anaerobic-Anoxic-Aerobic) to maintain the flow of mixed sewage and activated sludge, initial removal of suspended substances. Sewage samples are collected behind the oxidation ditch. |
| Inside the secondary sedimentation tank (D5, R5) | The flow velocity and amount of water affect the cross-section of rainwater, and the suspended matter rises. Sewage samples are collected in the secondary sedimentation tank. |
| After secondary settling tank (D6, R6) | The mud and water are separated and the suspended impurities form flocculates and sink together. Sewage samples are collected after the secondary sedimentation tank is selected. |
| Tail water or After filtration tank (D7, R7) | WWTP outflow tail water. Sewage samples are collected after the denitrification filter. |
| Dehydrated sludge (D8, R8) | After being dehydrated by an enrichment centrifuge. Sludge samples are collected on the conveyor belt. |

**Table 2.** Removal rate of MPs by tertiary treatment in WWTP.

| Treatment Phase | Arrange | MPs Abundance (n/L) | Removal Efficiency (%) | Total Removal Rate of MPs (%) |
|---|---|---|---|---|
| Primary processing | D1 to D3, R1 to R3 | $101.9 \pm 17.6$ to $51.0 \pm 7.3$, $108.7 \pm 20.1$ to $81.2 \pm 10.8$ | 62.9%, 70.4% | |
| Secondary treatment | D4 to D6, R4 to R6 | $71.9 \pm 15.3$ to $44.2 \pm 5.5$, $87.4 \pm 21.3$ to $53.6 \pm 7.4$ | 55.6%, 57.5% | 87.7% (D), 83.5% (R) |
| Tertiary or Advanced treatment | D6 to D7, R6 to R7 | $44.2 \pm 5.5$ to $18.2 \pm 3.6$, $53.6 \pm 7.4$ to $26.3 \pm 5.1$ | 44.9%, 34.6% | |

*2.3. Experimental Scheme*

2.3.1. MPs Separation and Extraction

In the Pretreatment Experiment (Figure 2), 1 L of sewage and 10 g of dry sludge were collected to assess the properties of MPs in various WWTP process structures. Coarse filtration was primarily performed on sewage samples utilizing stacked layers of stainless steel screens with pore sizes of 4 mesh (5 mm), 18 mesh (1 mm), 600 mesh (25 μm), and 1000 mesh (15 μm) (m1). The deionized water is then transferred to a sand core filter (JOAN LAB, 0.8 μm, 1000 mL, Huzhou, China) for filtration (m2). The coarse filtering screen is rinsed three times in deionized water. Following filtering, the filter membrane (PTFE, LONGJIN, aperture 5 μm, diam 50 mm, Nantong, China) was placed in a beaker containing 75 mL of Fenton reagent, exposed to a digestion reaction (m3), and permitted to stand for 12 h [41]. The residue left on the screen is still impinged on by a pressured water bottle carrying deionized water, which is subsequently filtered (m2) and the filter membrane is also immersed in the digesting solution (m3). To guarantee that the material is transmitted, the digested mixture is filtered (m4) and the Beaker is rinsed frequently with deionized

water after digestion. To initiate density separation [42], the digestion mixture is filtered through a membrane and combined (m5) with 300 mL of a flotation solution (saturated NaCl solution, likewise filtered by a filter). Beakers containing plastic microparticulate membranes were then put on a Magnetic stirrer (HUXI, HMS-203D, Shanghai, China) to expedite material removal from the membrane and shaken at 65 °C and 500 rpm for 24 h (m6). After density flotation, the supernatant was filtered (m7) and the residue in the Beaker was put back into the floatation solution for a second floatation (m6 and m7). This process was done three times. After three cycles of density separation, the three filter films are maintained in glass petri dishes, with the petri dishes coated in aluminium foil to prevent plastic particles from dispersing. For the treatment of the sludge sample, the sludge was positioned flat in a glass Petri dish, and then the Petri dish was placed in an oven where the sludge was dried at 105 °C for 24 h. The ensuing digestion procedures are identical to those for wastewater treatment, except that the mass of the sludge sample and the ratio of the input of the digestion solution are 1 g:30–50 mL (the dose of the digestion solution is determined by the removal rate of the digested material).

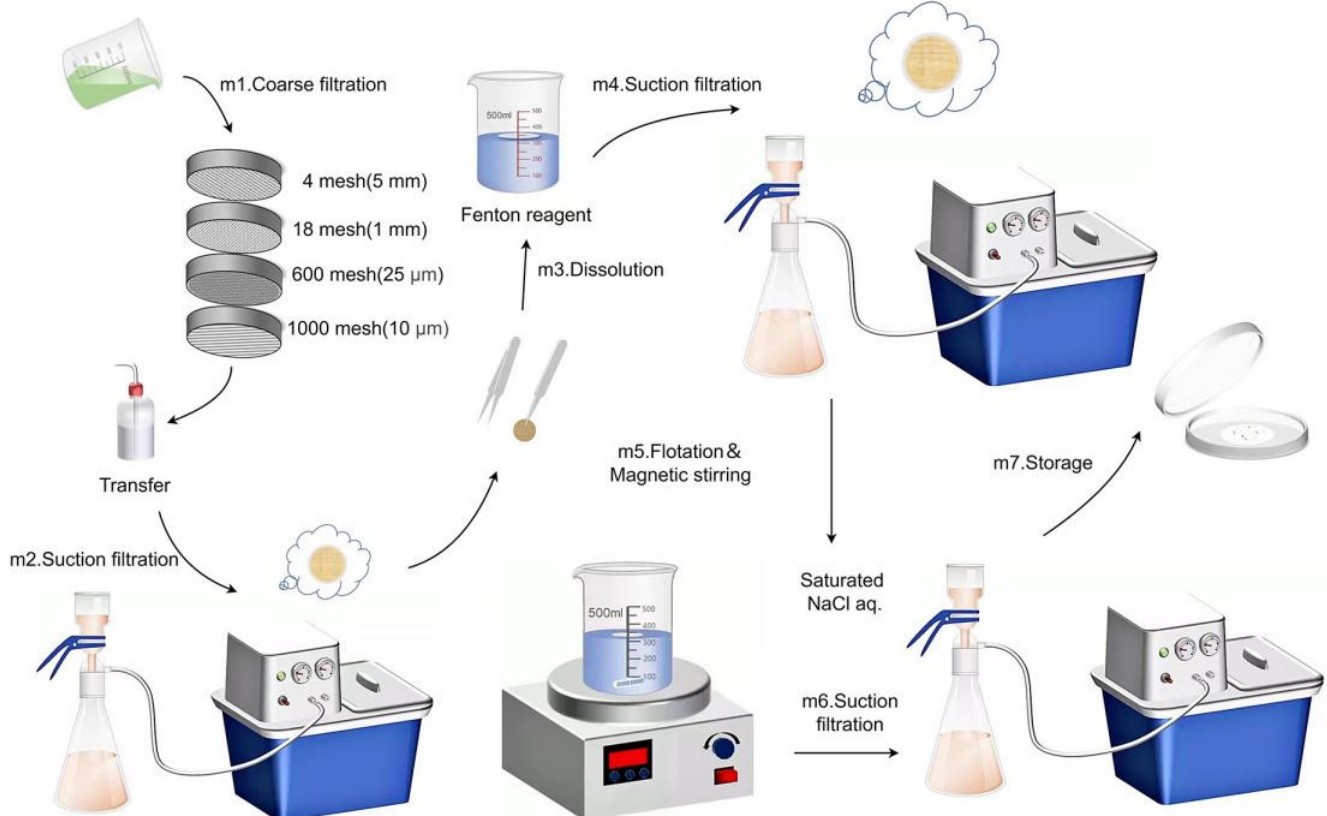

**Figure 2.** Experimental operation procedures.

2.3.2. Observation and Identification of Microplastics

Using an Osmosis electron microscope (AO-HK830-5870,Shenzhen, China), the morphological properties of putative MPs on a filter membrane were studied. Materials comprising plastic particles were dispersed over highly reflective glass in conjunction with micro-FTIR (Thermo Fisher Scientific Nicolet iN10, Waltham, MA, USA) in the region of 4000–400 cm$^{-1}$ with a spectral resolution of 4 cm$^{-1}$; An average of 64 scans were recorded [43]. KnowItAl soft-ware (BIORAD Inc., Hercules, CA, USA) was employed. The acquired spectra were compared with those from the Knowitall FTIR library (Bio-Rad Inc.) and the national standard of the People's Republic of China (GB/T 40146-2021, China) to define the polymer type of MP [44,45] based on the distinctive functional groups and peak trend rate. To explore the surface properties of MPs that can adsorb organic pollutants

such as polycyclic aromatic hydrocarbon, a scanning electron microscope (SEM, Hitachi S4801-IM, Japan) was used.

### 2.3.3. Quantitative Methods for Microplastics

For the characterization and quantification of MPs, reference is made to Pivokonsky et al. technique's of quantitative analysis of MPs by sampling 25% of the circular sector of each filter [46]. The five-point sampling approach was employed for MPs average abundance statistics. First, the central sample point is positioned at the midway of the diagonal on the high reflection glass (24 mm × 50 mm), and four points on the diagonal are determined as sampling sites. The region of interest (ROI) is selected by modifying the facula based on the mode of reflection. The area of the high-reflection glass lens comprises five squares: upper left (U1), upper right (U2), middle (C3), bottom left (D4) and lower right (D5); each quadrat is the same distance from the lens's centre (Tables 3 and 4) [47]. Then, the material on the lenses was infrared scanned individually based on 5 squares selected by a micro-FTIR (Thermo Fisher Scientific Nicolet iN10) surface scan, and the morphological traits and chemical composition of MPs were identified. After that, utilizing Formula (1), the total number of $N_m$ of MPs per litre of sewage or per gram of sludge collected throughout a single WWTP procedure is determined.

$$N_m = \frac{\sum_{i=0}^{5} * N_i * S_m}{5 S_f} \tag{1}$$

In Formula (1), $N_i$ is the number of MPs on each quadrate (n/L), $S_m$ is the contact area of impurities on a single high reflector, and $S_m \approx 9.26$ cm$^2$. $S_f$ is the area of a single quadrate, and $S_f = 0.84$ cm$^2$. The length of a quadrate side is 12.5 mm, and its breadth is 6.72 mm.

### 2.3.4. Experimental Quality Control

All containers and devices for collecting and storing MPs are composed of stainless steel or quartz glass. Clean the container several times with deionized water before sample and keep it sealed. Furthermore, using a pure cotton lab coat and nitrile gloves while sampling and experimenting is necessary to eliminate the shedding of fibers from textiles and clothing, which increases the exceptional amount of MPs. All of the containers are sealed with aluminum foil because, during the oxidation and exothermic digesting processes, plastic particles with increasing water vapour may be adsorbed on the foil, requiring immediate attention to avoid loss. MPs samples were collected in a confined clean room for microscopic and infrared spectroscopy examination.

### 2.3.5. Data Analysis

For data preprocessing and analysis, Microsoft Excel 2019 was used, SPSS 26.0 (IBM Co., Ltd., Armonk, NY, USA) for data correlation analysis, and Origin 2018 (OriginLab, Farmington, ME, USA) and Microsoft Visio 2016 for data analysis and charting. The abundance of MPs particles in this sewage treatment is expressed as the mean standard error. The MPs abundances of the Sewage Treatment samples collected during the three time periods were measured at a single process stage, and the average of the MPs abundances of the samples collected during the three time periods was taken as the range of final considerations for MPs in a single process stage. Furthermore, the data examined the MPS removal effectiveness of WWTP stages 1, 2, and 3, as well as the whole process from raw water to tail water. The findings demonstrate that: Removal Efficiency (%) is the removal efficiency of three stages, Total removal rate of MPs (%) is the total removal rate for the whole process stage. Sections 2.4 and 3.4, Tables 5 and 6 analyse the ecological risk coefficient and the assessment of pollution load contained in the MPs to evaluate the pollution risk of the MPs in the WWTP. Moreover, all data were examined for normality

(Shapiro-Wilk test) and variance homogeneity (Levene test). The statistical significance threshold was established at $p < 0.05$.

### 2.4. Potential Ecological Risk Assessment of MPs

In order to restrict the spread of MPs pollution, it is necessary to quantify the potential ecological danger posed by MPs pollution. In this paper, the potential ecological risk of MPs in WWTP is evaluated using the MPS pollution load index (PLI) model, which was initially developed to evaluate the level of water pollution in estuaries; it is now expanded to calculate the value-at-risk of MPs [48]. The following is the formula:

$$CF_i = C_i/C_{oi} \tag{2}$$

In Formula (2), $CF_i$ is defined as the ratio of MPs abundance ($C_i$) at each sampling point to MPs minimum abundance ($C_{oi}$) at each sampling point.

$$PLI_i = \sqrt{CF_i} \tag{3}$$

$$PLI_{zone} = \sqrt[n]{PLI_1 \times PLI_2 \times \ldots \times PLI_n} \tag{4}$$

In Formula (3) and (4), $PLI_i$ represents the pollution load index of MPs for a single sample, whereas $PLI_{zone}$ represents the pollution load index of MPs for WWTP.

$$H = \sum P_n \times S_n \tag{5}$$

In Formula (5), $H$ is the MPs potential ecological risk index, the proportion of each MPs polymer type at each sampling site for the $P_n$, and $S_n$ is the hazard score for the sample point MPs polymer [49] (Tables 5 and 6).

**Table 3.** MPs fixed-point quantification (WWTP in dry weather, Sewage (n/L), Sludge (n/10 g)).

| Process Segment | D1 | D2 | D3 | D4 | D5 | D6 | D7 | D8 |
|---|---|---|---|---|---|---|---|---|
| Scanning points (U1, U2, C3, D4, D5) | 125 | 93 | 62 | 137 | 104 | 78 | 34 | 260 |
| Shape (Formula (1)) | | | | | | | | |
| Fiber | 5.9 | 1.7 | 1.9 | 4.3 | 1.7 | 2.0 | 0.9 | 5.6 |
| Chip | 2.8 | 1.3 | 1.2 | 1.1 | 1.3 | 0.9 | 0.3 | 3.9 |
| Sheet | 1.6 | 1.4 | 0.5 | 1.0 | 0.9 | 0.5 | 0.1 | 2.5 |
| Particle | 0.9 | 0.1 | 0.2 | 0.5 | 0.2 | 0 | 0 | 1.5 |
| Size (Formula (1)) | | | | | | | | |
| 0–100 µm | 2.8 | 1.8 | 1.4 | 3.0 | 1.4 | 1.1 | 0.7 | 4.9 |
| 100–500 µm | 4.1 | 1.0 | 0.8 | 2.4 | 1.6 | 1.8 | 0.5 | 2.3 |
| 500–1000 µm | 2.0 | 1.2 | 1.3 | 1.1 | 0.5 | 0.4 | 0.1 | 3.0 |
| 1000–2500 µm | 1.7 | 0.6 | 0.3 | 0.4 | 0.7 | 0.1 | 0 | 1.7 |
| 2500–5000 µm | 0.6 | 0 | 0 | 0.1 | 0.1 | 0 | 0 | 1.5 |
| Actual MPs abundance, Formula (1) | $101.9 \pm 17.6$ | $61.1 \pm 9.3$ | $51.0 \pm 7.3$ | $71.9 \pm 15.3$ | $68.1 \pm 13.6$ | $44.2 \pm 5.5$ | $18.2 \pm 3.6$ | $184.8 \pm 28.6$ |

**Table 4.** MPs fixed-point quantification (WWTP in rainy weather, Sewage (n/L), Sludge (n/ 10 g)).

| Process Segment | R1 | R2 | R3 | R4 | R5 | R6 | R7 | R8 |
|---|---|---|---|---|---|---|---|---|
| Scanning points, (U1, U2, C3, D4, D5) | 85 | 93 | 82 | 113 | 131 | 57 | 55 | 277 |
| Shape, (Formula (1)) | | | | | | | | |
| Fiber | 5.0 | 2.3 | 1.9 | 2.3 | 3.8 | 2.8 | 1.7 | 5.5 |
| Chip | 2.8 | 1.6 | 1.2 | 0.1 | 1.0 | 0.9 | 0.1 | 3.0 |
| Sheet | 1.8 | 2.0 | 0.5 | 1.0 | 1.0 | 0.7 | 0.2 | 2.5 |
| Particle | 1.1 | 0.3 | 0.2 | 0 | 0.5 | 0 | 0 | 1.5 |
| Size, (Formula (1)) | | | | | | | | |
| 0–100 μm | 1.7 | 2.1 | 1.9 | 1.9 | 3.3 | 1.5 | 1.0 | 3.3 |
| 100–500 μm | 4.0 | 1.8 | 1.3 | 1.4 | 3.7 | 1.8 | 0.8 | 5.2 |
| 500–1000 μm | 1.3 | 1.4 | 0.6 | 2.4 | 1.4 | 0.5 | 0.1 | 2.1 |
| 1000–2500 μm | 1.7 | 0.5 | 0.3 | 0.5 | 0.3 | 0.6 | 0 | 0.9 |
| 2500–5000 μm | 1.1 | 0.4 | 0 | 0.1 | 0.1 | 0 | 0 | 1.1 |
| Actual MPs abundance, Formula (1) | 108.7 ± 20.1 | 77.9 ± 11 | 81.2 ± 10.8 | 87.4 ± 21.3 | 117.3 ± 22.4 | 53.6 ± 7.4 | 26.3 ± 5.1 | 178.4 ± 34.3 |

**Table 5.** Risk evaluation of MPs in WWTP (1).

| Type of Polymer | PE | | PP | | PS | | PET | |
|---|---|---|---|---|---|---|---|---|
| Hazard score (Highest level); $S_n$ | 11 | | 1 | | 4 | | 30 | |
| Process Segment | D | R | D | R | D | R | D | R |
| $P_n$ (%) | 11.30 | 15.89 | 6.88 | 12.33 | 9.83 | 8.60 | 10.81 | 8.60 |
| $H$, (Formula (5)) | 1.24 | 1.75 | 0.07 | 0.12 | 0.39 | 0.34 | 3.24 | 2.58 |
| Potential ecplogical risk level of MPs | I | I | I | I | I | I | I | I |
| $PLI_i$ (Formula (2) and (3)) | 2.40 | 2.62 | 1.88 | 2.30 | 1.93 | 2.72 | 1.63 | 1.95 |

**Table 6.** Risk evaluation of MPs in WWTP (2).

| Type of Polymer | PU | | PA | | PF | | PVC | |
|---|---|---|---|---|---|---|---|---|
| Hazard score (Highest level); $S_n$ | 871 | | 50 | | 1450 | | 30 | |
| Process Segment | D | R | D | R | D | R | D | R |
| $P_n$ (%) | 5.65 | 6.54 | 24.32 | 21.68 | 17.94 | 14.95 | 13.27 | 11.40 |
| $H$, (Formula (5)) | 49.22 | 56.98 | 352.70 | 314.39 | 1324.40 | 1104.15 | 663.52 | 570.20 |
| Potential ecplogical risk level of MPs | II | II | III | III | III | III | III | III |
| $PLI_i$ (Formula (2) and (3)) | 1.00 | 1.51 | 1.04 | 1.00 | 1.73 | 2.39 | 1.97 | 2.42 |
| $PLI_{zone}$ (Formula (4)) | $PLI_{zone}$ (Dry weather) value is 1.63 (moderately pollution), $PLI_{zone}$ (Rainy weather) value is 2.03 (highly pollution) | | | | | | | |

## 3. Results and Discussion

### 3.1. Distribution and Reduction of MPS in WWTP

In this investigation, 14 wastewater samples were collected from different typical WWTP stages (D1 to D8, R1 to R8) under dry (D) and rainy (R) weather conditions, together with two samples of dewatered sludge. These samples are used to illustrate the distribution and fluctuation of MPs following Sewage Treatment treatment at different phases of the procedure. Following are the abundance values of MPs at the WWTP under dry and rainy

circumstances. The abundance of MPs at D1 to D8 was $101.9 \pm 17.6$, $61.1 \pm 9.3$, $51.0 \pm 7.3$, $71.9 \pm 15.3$, $68.1 \pm 13.6$, $44.2 \pm 5.5$, $18.2 \pm 3.6$ n/L, $184.8 \pm 28.6$ n/10 g. The abundance of MPs was $108.7 \pm 20.1$, $77.9 \pm 11.0$, $81.2 \pm 10.8$, $87.4 \pm 21.3$, $117.3 \pm 22.4$, $53.6 \pm 7.4$, $26.3 \pm 5.1$ n/L, $178.4 \pm 34.3$ n/10 gat the R1 to R8 sampling sites, respectively (Tables 3 and 4).

Figure 3 depicts the overall distribution of MPs in the typical WWTP process phases, in which rainy conditions fluctuate significantly. The wastewater at R4 (4. After oxidation Ditch) and R5 (5. Inside the secondary sedimentation tank) was somewhat more turbid than that at D4 and D5, and the quantity of MPs at R5 increased by 19.8% as compared to D5. This is mostly attributable to the overflowing of sewage in the sewers, the rise in the treatment load on process equipment, and the increase in flow disruption when precipitation enters the Sewage Treatment. The impact of reflux in R4 was diminished, the settling time in R5 was shortened, and the performance of the activated sludge system was diminished [50]. These variables contribute to the difference between R5 and D5 MPs.

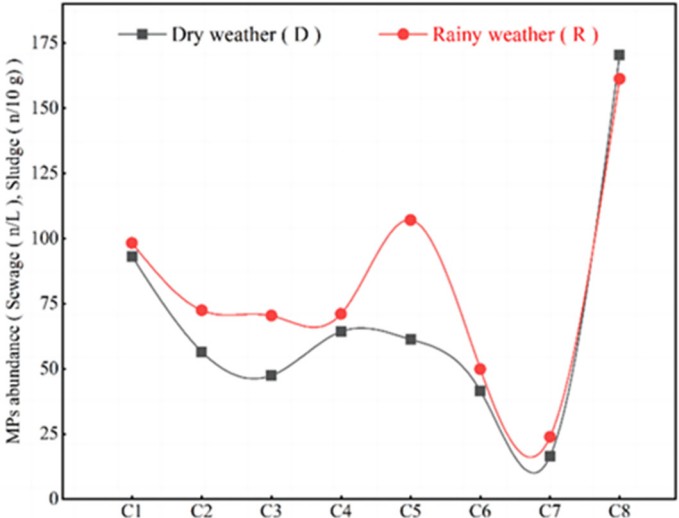

**Figure 3.** Comparison of the MPs variation trend for dry and rainy weather in typical process stage of WWTP. C: Craft (C1 to C8).

MPs abundance significantly decreased in the first stage of the WWTP, C1 to C3 (1. Row water, 2. After grille, 3. After sedimentation Tank), at rates of 33.3% and 25.0%, respectively (Figure 3). The interception of the grid and the role of flocculation and sedimentation in the grit chamber, which effectively precipitated suspended and colloidal material with a size of less than 100 μm in the grit chamber, were potential causes for the decline [51–53]. Table 2 demonstrates that the rates of primary removal were 62.9% and 70.4%, respectively. Nonetheless, the MP's D4 and R4 effluent abundances increased by 51.7% and 35.2%, respectively. Refluxing of the aeration tube in the aerobic portion modifies the structure of MPs, making it simpler for large plastics to be degraded by anaerobic processes, while breaking down into smaller plastic particles [54,55]. Compared to the D4 and R4 stages, the D5–D6 and R5–R6 (5. Inside the secondary sedimentation tank to 6. After the secondary settling tank) stages were partially settled by MPs. The removal rates of subsequent therapy were 55.6% and 57.5%, respectively. Tertiary treatment of filter sedimentation and disinfection only decreased contamination of treated water and chemical contamination indicators [56], with little influence on changes in MPs abundance. In three stages, the removal rates were 44.9% and 34.6%, respectively. The overall removal rates for raw water and discharge water were 87.7% and 83.5%, respectively. In addition, the MPs present in the sludge under both weather conditions were $184.8 \pm 28.6$ n/10 g, $178.4 \pm 34.3$ n/10 g. Compared to 79 sludge samples taken from 28 Chinese Sewage Treatment Facilities, the estimated average quantity of MPs in Sewage Treatment Sludge was $22.7 \pm 12.1$ n/g. MPs abundance significantly decreased in the first stage of the WWTP, C1 to C3 (1. Row water, 2. After grille, 3. After sedimentation Tank), at rates of 33.3% and 25.0%, respectively

(Figure 3). The interception of the grid and the role of flocculation and sedimentation in the grit chamber, which effectively precipitated suspended and colloidal material with a size of less than 100 μm in the grit chamber, were potential causes for the decline. Table 2 demonstrates that the rates of primary removal were 62.9% and 70.4%, respectively. Nonetheless, the MP's D4 and R4 effluent abundances increased by 51.7% and 35.2%, respectively. Refluxing of the aeration tube in the aerobic portion modifies the structure of MPs, making it simpler for large plastics to be degraded by anaerobic processes, while breaking down into smaller plastic particles [54,55]. Compared to the D4 and R4 stages, the D5–D6 and R5–R6(5. Inside the secondary sedimentation tank to 6. After the secondary settling tank) stages were partially settled by MPs. The removal rates of subsequent therapy were 55.6% and 57.5%, respectively. Tertiary treatment of filter sedimentation and disinfection only decreased contamination of treated water and chemical contamination indicators [56], with little influence on changes in MPs abundance. In the three stages, the removal rates were 44.9% and 34.6%, respectively. The overall removal rates for raw water and discharge water were 87.7% and 83.5%, respectively. In addition, the MPs present in the sludge under both weather conditions were 184.8 ± 28.6 n/10 g, 178.4 ± 34.3 n/10 g. Compared to 79 sludge samples taken from 28 Chinese Sewage Treatment facilities, the estimated average quantity of MPs in Sewage Treatment sludge was 22.7 ± 12.1 n/g.

### 3.2. Source and Variation of MPs in Different Polymer Types in WWTP

The aggregate proportion of various MPs types in wastewater and sludge samples at typical WWTP stages in dry and rainy weather conditions was determined using micro-FTIR analysis (Tables 5 and 6). The plastics examined included PE (Polyethylene), PP (Polypropylene), PA (Polyamid), PVC (Polyvinylchloride), PS (Polystyrene), PET (PolyethyleneTerephthalate), PU (Polyurethane), PF (phenol-formaldehyde resin), which are the primary types of polymers that can be detected in the ordinary phases of a WWTP process in both rainy and dry weather, non-plastic is the impurity that interferes with the membrane and a non-plastic polymer. Figure 4 demonstrates that the average composition of the three plastic polymers PA, PF, and PE is greater than that of other polymers. In dry weather, the proportion of PA polymer was 24.32% and in rainy weather, it was 21.68%, followed by PF at 17.94% and PE at 11.30% and 15.89%. In the initial stage of physical treatment, the fine nylon fiber's PA interception effectiveness is limited. However, during the oxidation ditch and secondary sedimentation phases, enrichment is observed. This might be the secondary treatment (4. Oxidation ditch) stage because of the fluidity of the activated sludge, which results in the capture of the majority of polymers, including PA, at this point [51]. PE, PS and PA are frequent textile materials that are frequently derived from laundry. At the same time, the source of PE and PS can be used in personal care products as abrasive particles and in cosmetics as an absorbent. Most likely, PF and PU sources come from the wide use of electrical insulation and rubberized fabric. This is because an electrical processing factory is close to the WWTP. PVC and PET can be used in plastic greenhouses on agricultural land and come from various suppliers. The films are naturally aged and photocatalytic, resulting in debris and films that are washed away by rainfall, which will be dispersed in the plastic surface environment and transported to the municipal WWTP [57,58]. Based on how plastic polymer parts are spread out in the typical phases of the WWTP process, residential sewage is the main source of MPs. Rainfall is an outside factor that contributes to the spread of MPs.

### 3.3. Shape, Distribution and Size of MPs in Sewage Plants

Figures 5 and 6 illustrate the shape and size properties of MPs in sewage and sludge at various WWPT processes. The MPs was separated into four categories: fiber, chip, sheet, and particle. The dimensions are separated into five categories (Figure 7): 0–100, 100–500, 500–1000, 1000–2500, 2500–5000 μm. Figures 8 and 9 show that fibrous MPs in sewage and sludge are most numerous at each typical WWTP process stage, with 49.3% and 39.7% of sewage and sludge abundances in dry weather, respectively, and 50.1% and

43.2% of sewage and sludge abundances in rainy weather, respectively. Following that are particle, fragment, and film shapes. It can be established that the abundance of MPs in various forms is proportional to the source and composition of sewage, as well as the rate of change in sewage volume and pollutant content. Fibrous MPs have been discovered in sewage and sludge, primarily from washing-worn fabric fibres and synthetic fibres floating in the air [59–61]. Fragments with a particle size of smaller than 5 mm are more likely to have originated from plastic products as a result of environmental elements such as light, thermal oxidation, and physical friction during use or disposal [62]. The artificial generation of abrasives in industrial production and the consumption of personal care products and cosmetics containing significant quantities of plastic bead particles are the origins of the particle shape [4,63]. Figure 10 reveals that the source of the film is due in part to the usage of plastic bags by residents and in part to the extensive use of plastic greenhouses on agricultural land in the upper Sewage Treatment Basin. Direct sunlight and extreme precipitation, such as snow, and hail, expedited the deterioration of plastic green-houses, leaving shattered film residue on the soil and agricultural products [64].

Figures 8 and 9 demonstrate that MPs in the 0–500 μm size range had the greatest average distribution in WWTP, with sewage and sludge abundances of 64.9% and 60.4% in dry weather and 67.9% and 69.0% in rainy weather, respectively. The second is 500–1000 μm, or more than 1000 μm of plastic. The results revealed that the sludge was tightly concentrated with MPs fragments and fibers [65], with a microparticle-sized plastics content below 1000 μm ranging from 88.3–91.2%. During mud cake processing, MPs are fragmented into smaller plastic particles, thereby boosting MPs' abundance [66]. MPs with high density and large size may have been trapped by sedimentation in the region of 2500–5000 μm (1. Row water to 5. Inside the secondary sedimentation tank) and (6. After secondary settling tank). However, the fraction of MPs with a size of 0–500 μm expanded from 24.8% to 42.6%, most probably as a result of the aeration and activated sludge treatment process (4. After oxidation Ditch), which breaks down large plastics into smaller plastic particles [55,67]. The proportion of MPs with particle sizes ranging from 500–1000 μm remained steady, ranging between 11.2% and 18.8%. It demonstrates that larger particle-size MPs can be efficiently eliminated during the first and second treatment phases, but smaller MPs can be removed slower.

### 3.4. Contamination Risk Evaluation of MPs in WWTP

Sewage and sludge from WWTP are processed in various ways and still include MPs at each step. The lower MPs, along with the tail water and sludge from the dewatering pump house, will be dumped into the river's natural flow and could be used as fertilizer for farming and urban greening. These approaches, on the one hand, augment the pollutant load index of MPs at WWTP. On the other hand, MP migratory behaviour in the natural environment may provide an acute or chronic risk to the ecology, either directly or indirectly [68–70]. Firstly, when dividing the risk series of MPs in WWTP and its associated toxicity, the characteristics of MPs in each typical process stage are evaluated, and the European classification, labelling, and packaging (CLP) standard is referred to. The risk series is then divided into four grades, and the pollution load is divided into three types. The model was based on the risk index (H), and the danger levels varied from (I) (<10 slightly toxic) through (II) (10–100 moderately toxic), (III) (100–1000 highly toxic), and (IV) (>1000 very highly toxic). Each level is given an approximate risk rating, with each danger level (I–IV) rising tenfold. It was picked ten times because it differentiates between various degrees of toxicity risk. It is also a unique categorization criterion used in GHS to differentiate between acute and chronic risk categories in the aquatic environment [49]. Secondly, the three categories of pollution loads were determined using the pollution coefficients of the MPs pollution load index (PLIi) in a single sample and the MPs pollution load index (PLIzone) in the total research area: (<1 slightly pollution, 1–2 moderately pollution, >2 highly pollution). The pollution risk of MPs was eventually determined by combining it with the MPs risk index in WWTP [71].

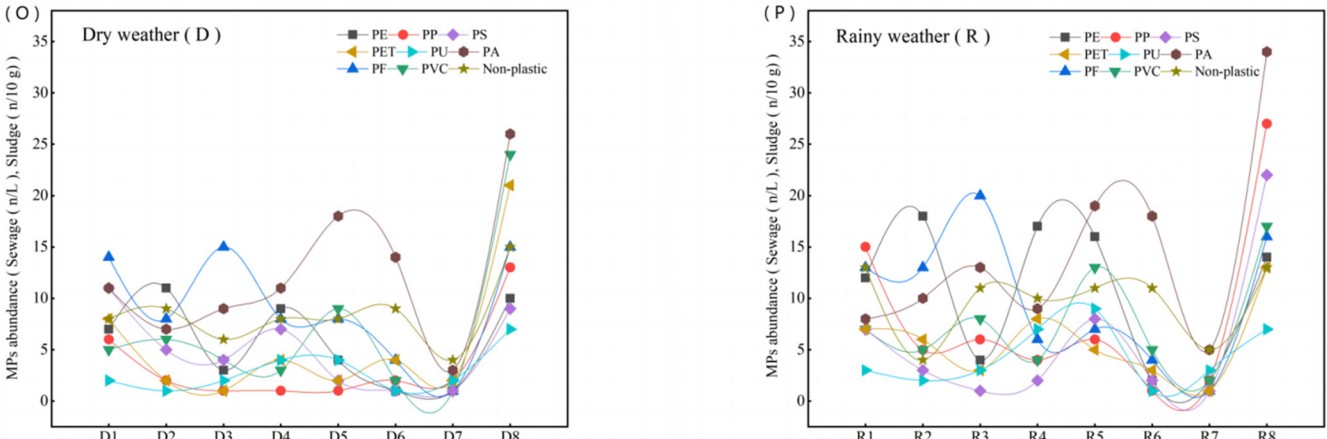

**Figure 4.** Variation of various MPs polymers in dry and rainy weather during a typical WWTP process stage. (**O**): Polymer in dry weather; (**P**): Polymer in rainy weather.

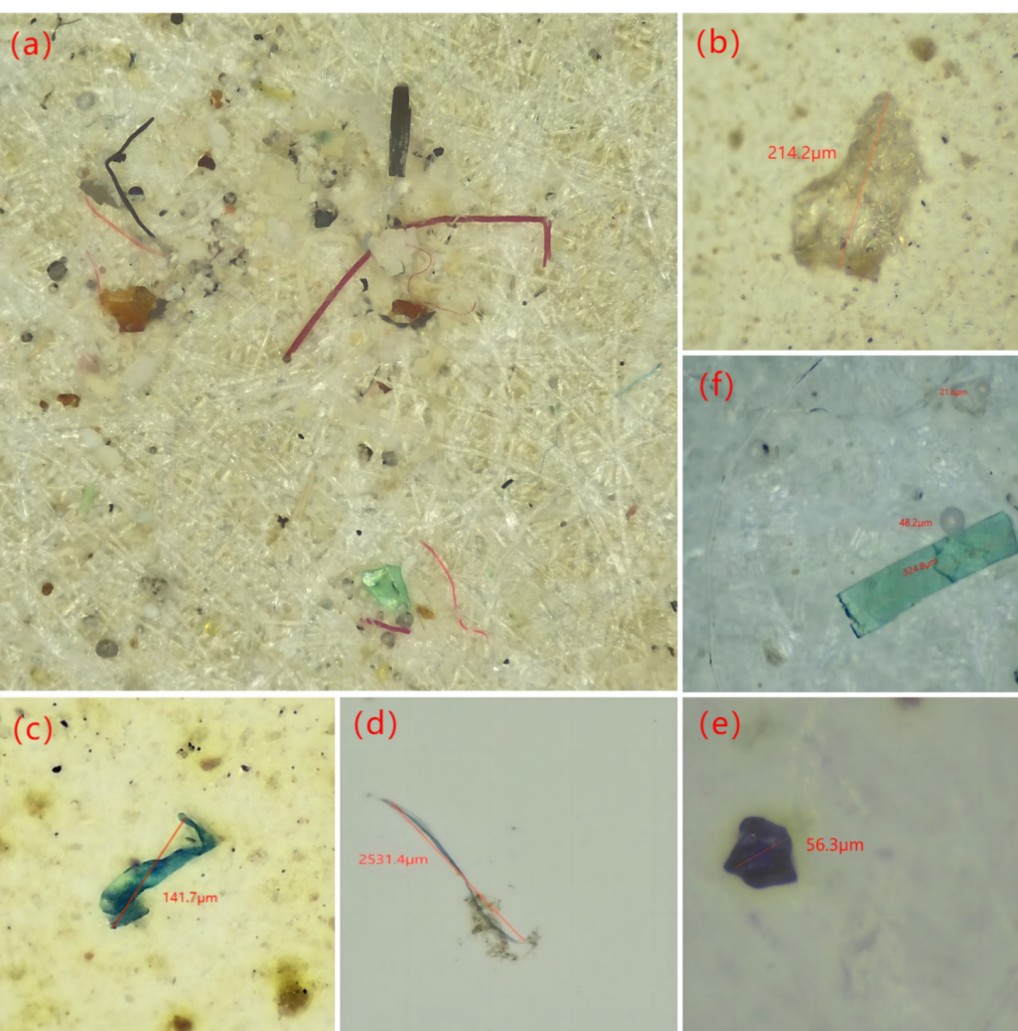

**Figure 5.** Multiple shapes of MPs identified in WWTP. (**a**): MPs in quadrat; (**b**,**c**): Chip; (**d**): Fiber; (**e**): Sheet; (**f**): Chip, Particle.

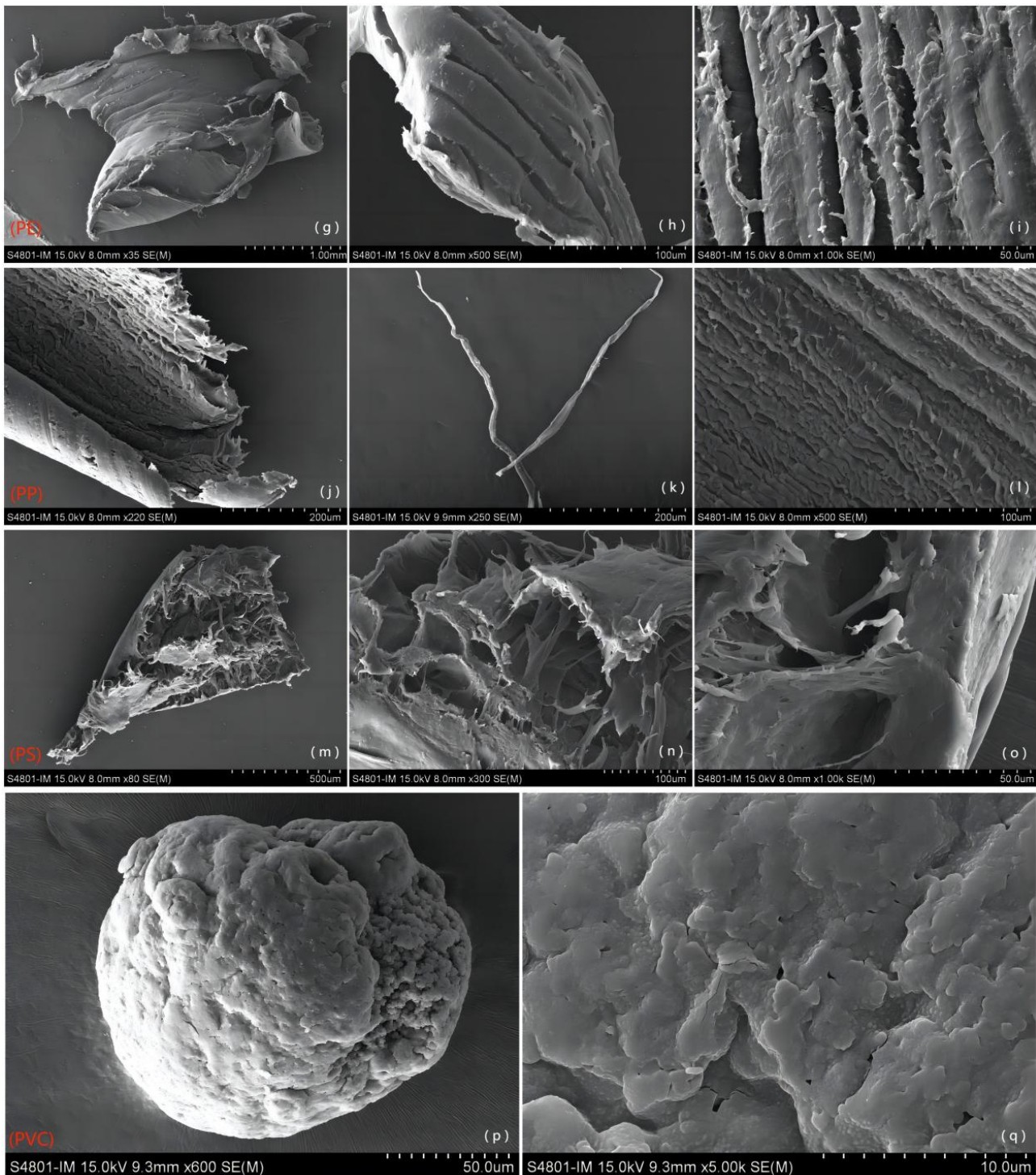

**Figure 6.** The surface-characteristic morphology of MPs. (**g**–**i**) are (PE); (**j**–**l**) are (PP); (**m**–**o**) are (PS); (**p**,**q**) is (PVC).

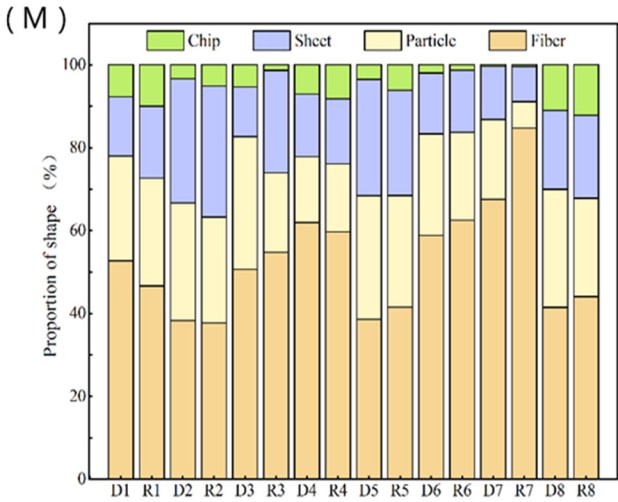

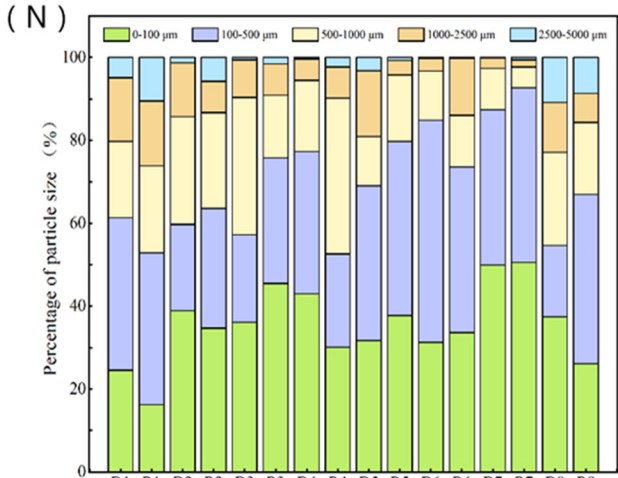

**Figure 7.** Characteristics of MPs in the sewage and sludge of the WWTP. (**M**): Shape of MPs, (**N**): Size of MPs.

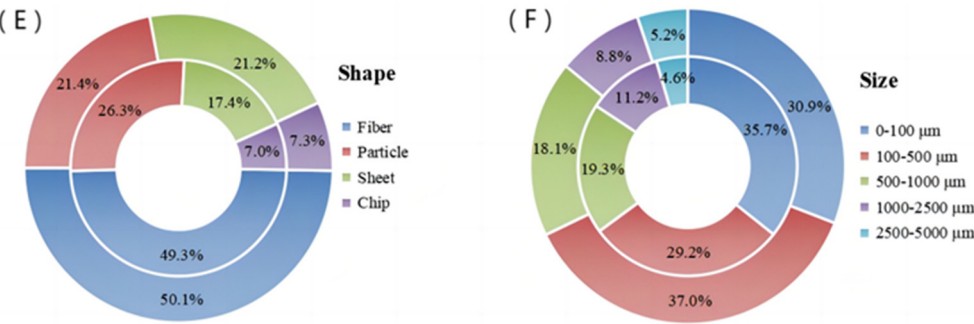

**Figure 8.** The aggregate relative proportions of MPs of varying shapes and sizes in sewage at the WWTP. (**E**): Shape, (**F**): Size; Dry and rainy weather conditions are represented by the inner and outer rings.

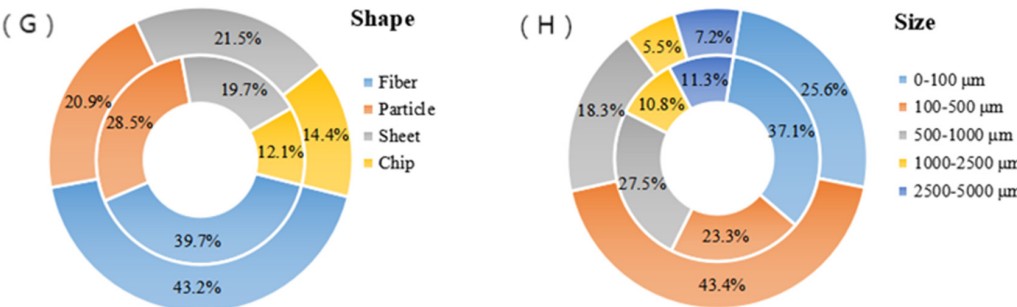

**Figure 9.** The overall relative proportions of MPs of different shapes and sizes in sludge at the WWTP. (**G**): Shape, (**H**): Size.

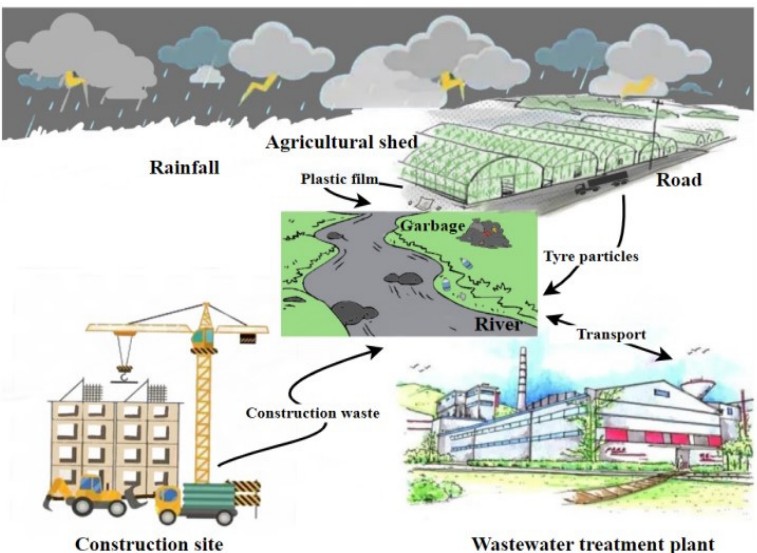

**Figure 10.** Migration of MPs in the freshwater environment.

Under both rainy and dry weather circumstances, the pollution index of MPs in Row water is 2.40 and 2.46, respectively, according to formula (5), indicating that the WWTP is significantly polluted. The tail water pollution index ranged from 1.0 to 1.2, suggesting considerable contamination. It demonstrates that the current procedure influences the expulsion of MPs. However, MPs in dehydrated sludge had a pollution index of 3.5 under dry weather circumstances and 3.4 under rainy weather conditions, indicating that they were highly polluting. The damage caused by sludge seems more substantial than that generated by sewage release. It is recommended that this substance be incinerated, since it is not suited for agricultural or urban greening fertilizer. Meanwhile, Tables 5 and 6 demonstrate that in dry and rainy conditions, the risk levels of PE, PP, PS PET polymer are (I), PU and PA are (II), and PVC and PF polymer are (III). The abundance of MPs $18.2 \pm 3.6$ n/L at sample point D7 was employed to calculate the $C_{oi}$ for this investigation [47]. The $PLI_i$ values of PP, PU, PVC, PS, PET, PA and PF in the typical process stages of WWTP are in the range of (1–2 moderately pollution) in dry weather (D1–D8), while PE polymers are in the (>2 highly pollution) range. The $PLI_i$ values for PET, PA, and PF were in the (1–2 moderately pollution) range under rainy circumstances (R1–R8), whereas those for PE, PP, PU, PVC, and PS were in the (>2 highly pollution) range. Ultimately, the $PLI_{zone}$ Index of the total MPs in the WWTP was determined using the formula (4). In dry weather (D1–D8), the $PLI_{zone}$ value was 1.63, which was moderately polluted. In comparison, the $PLI_{zone}$ value is 2.03, which is extremely polluted. Collectively, the percentage of different kinds of polymers and their hazard ratings were strongly connected with the pollution risk of MPs in the WWTP; the MPS Pollution Index was more volatile in rainy weather. The primary source of its effects is probably sewage sources, with non-point source migration

of pollution sources, including plastics, into the WWTP owing to rainfall, exacerbating the pollution risk of MPs in the WWTP [72].

## 4. Conclusions

In this paper, MPs abundances in sewage diminished most dramatically following primary and secondary treatment utilizing the WWTP, with average removal ratios of 59.3% and 64.0% during dry and rainy weather, respectively. However, sedimentation and disinfection from the three-stage treatment lessened the pollutant and Chemical Pollution Index further, and the MPs elimination effectiveness was only 44.9% and 34.6%, respectively. 87.7% and 83.5% of all MPs were terminated. Sewage and sludge from WWTPs had the most incredible average abundance of fibrous MPs, 49.3% and 39.7% in dry weather and 50.1% and 43.2% in rainy weather, respectively. With 64.9% and 60.4% in dry weather and 67.9% and 69.0% in rainy weather, respectively, WWTP wastewater and sludge had the greatest average distribution of MPs in the range of 0–500 μm. It demonstrates that the form and size of MPs are changing as a result of the WWTP process, and that sludge accumulation is considerable.

According to a micro-FTIR spectrometer, the predominant constituents of WWTP MPs include PP, PE, PS, PA, PET, PU, PF and PVC. It has been discovered that the types of polymers in question are strongly linked to human activities. The primary source of these polymers is sewage from homes, which can be affected by weather conditions like rain. Also, the pollution risk of MPs in the WWTP was related to the amount of polymers and their hazard scores both when it was dry and when it was raining The abundances of MPs in the WWTP's Row water were 101.9 ± 17.6 n/L and 108.7 ± 20.1 n/L, respectively. Extremely high pollution risk indices of 2.40 and 2.46 were calculated using the PLI pollution load index model. The concentrations of MPs in Tail water were 18.2 ± 3.6 n/L and 26.3 ± 5.1 n/L, with corresponding pollution risk indices of 1.0 and 1.2. The findings indicate that the current WWTP procedure influences the elimination of MPs. The abundances of MPs in dewatered sludge were 184.8 ± 28.6 n/10 g and 178.4 ± 34.3 n/10 g, and their pollution indices were 3.5 and 3.4, which were both extremely polluted. Consequently, there is a potential danger of ecological contamination since there is still an outflow of MPs from the WWTP process before and after treatment, with the tail water being discharged into the natural water body or deposited in the sludge, causing the movement of MPs in the environment. The interception and removal effectiveness of MPs need to be enhanced.

**Author Contributions:** Writing—original draft preparation, X.M.; conceptualization, writing—review and editing, T.B.; formal analysis, data curation, L.H.; supervision, project administration, K.W. All authors have read and agreed to the published version of the manuscript.

**Funding:** This research was supported by the National Key R&D Program of China (2020YFC1908601, 2020YFC1908602); National Special Item on Water Resource and Environment (2017ZX07603-003); Hefei University's major projects research grant (Grant No. 20RC42).

**Conflicts of Interest:** The authors declare no conflict of interest.

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
