# Peer review of "Occurrence Characterization and Contamination Risk Evaluation of Microplastics in Hefei’s Urban Wastewater Treatment Plant"

_water, doi:10.3390/w15040686_

Round 1
Reviewer 1 Report
The paper focused on the evaluation of MPs pollution risk in urban WWTPs in Hefei City , and investigated the form, size, and composition features of MPs in each typical process step of a WWTP under two weather conditions, dry and rain, as well as the removal effectiveness of MPs in the three-stage treatment stage, which serves as a reference for future MPs reductions in China's inland cities' WWTP.
Overall, the work of this paper is practical and logical, which were helpful to serve as a reference for future MPs reductions in China's inland cities' WWTP.
Specific comments:
“2.1. Sample sites”: Please add the specific location map of the sampling point.
“2.3.1. MPs separation and extraction”: In this part, the separation and extraction method of MPs is proposed. what is the recovery rate of this method and what is the repeatability of the experiment? It is suggested that the author give.
“Line357-359”: The paper mentions the primary source of MPs is residential sewage. In addition to the references, there is no sewage discharge data near the sampling sites, which can support the conclusion. Recommendation description.
“4. Conclusion”: The conclusion should be written on a standalone basis highlighting the significant findings. The recommended conclusions are listed item by item.
Overall Recommendation: Minor Revision.

Reviewer 2 Report
The article entitled "Occurrence characterization and contamination risk evaluation of microplastics in Hefei’s urban wastewater treatment plant" by Meng et al. is of high quality and originality and should be considered for publication in Water, MDPI. The authors have put a significant effort to address a major problem in Hefei's urban wastewater treatment plant in China, by critical and in-depth analysis of the different types of Microplastics (MPs) at different environmental conditions. I enjoyed reading this manuscript and I strongly recommend this work for publication, however the English language should be carefully revised. Also all abbreviations in the abstract should be clearly defined before being abbreviated.
Reviewer 3 Report
It is practical studies then reflecting the impact of microplastic to the receiving water bodies from effluent of wastewater treatment plants. the manuscript was worth publishing after some questions were clarified. First of all, are there any data from other wastewater plant can be compared with this study? are there any variation between the MPs distribution of Hefei city and the nearby city? Are there any method validation of the experimental operation procedure and hence the % recovery of the method? In the part of experimental quality control, link 224, it stated that the plastic particles with increasing water vapor may be absorbed on the aluminum foil, are there a reference supported? IN figure 4, it is difficult to examine the trend of each type of polymer variation, it is suggested to change the presenting form. Lastly, what is the impact of MP’s shape to the ecological risk? It may need to explain more in the discussion parts.
Reviewer 4 Report
This work investigated the form, size, and com-position features of MPs in each typical process step of a WWTP under two weather conditions, dry and rain, as well as the removal effectiveness of MPs in the three-stage treatment stage. Some comments for improving the manuscript are given below.
1. 2. Materials and experimental methods: Many contents in this part have been reported in other articles for many times. I suggest the author simplify the introduction.
2. The references in the manuscript are too old to represent the latest research progress in this field.
3. The reviewer suggested that the author carefully revise the format of the manuscript according to the author's guide. The current format cannot meet the publishing requirements.
4. The following literature may be helpful in improving the quality of the manuscript. (Journal of Hazardous Materials, 2022, 426: 128062. ACS Applied Materials & Interfaces, 2022, 14(5): 7450-7463.)
5. Highlight: The journal does not need to provide Highlights.
6. The author should highlight the relationship between the manuscript and the subject of the journal.
Round 2
Reviewer 4 Report
After the first run of revision, the manuscript improved in quality. However, I will recommend it be published until the following comments are addressed.
1. Compared with other works, what is the innovation of this article?
2. Figure 6: The picture layout is too crowded. As shown in Figure 5, there should be gaps between each figure.
3. The author should simplify the introduction of the formula, because this part has been reported in many literatures.
